# Analysis of allelic cross-reactivity of monoclonal IgG antibodies by a multiplexed reverse FluoroSpot assay

**Henriette Hoffmann-Veltung[1], Nsoh Godwin Anabire[1,2,3], Michael Fokuo Ofori[3], Peter Janhmatz[4], Niklas Ahlborg[4,5], Lars Hviid[1,6], Maria del Pilar Quintana[1]***

[1]Centre for Medical Parasitology, Department of Immunology and Microbiology, Faculty of Health and Medical Sciences, University of Copenhagen, Copenhagen, Denmark; [2]West African Centre for Cell Biology of Infectious Pathogens, Department of Biochemistry, Cell and Molecular Biology, University of Ghana, Accra, Ghana; [3]Department of Immunology, Noguchi Memorial Institute for Medical Research, University of Ghana, Accra, Ghana; [4]Mabtech AB, Nacka Strand, Sweden; [5]Department of Molecular Biosciences, The Wenner-Gren Institute, Stockholm, Sweden; [6]Department of Infectious Diseases, Rigshospitalet, Copenhagen, Denmark

*For correspondence:
pilar@sund.ku.dk

**Abstract** The issue of antibody cross-reactivity is of central importance in immunology, and not least in protective immunity to *Plasmodium falciparum* malaria, where key antigens show substantial allelic variation (polymorphism). However, serological analysis often does not allow the distinction between true cross-reactivity (one antibody recognizing multiple antigen variants) and apparent cross-reactivity (presence of multiple variant-specific antibodies), as it requires analysis at the single B-cell/monoclonal antibody level. ELISpot is an assay that enables that, and a recently developed multiplexed variant of ELISpot (FluoroSpot) facilitates simultaneous assessment of B-cell/antibody reactivity to several different antigens. In this study, we present a further enhancement of this assay that makes direct analysis of monoclonal antibody-level cross-reactivity with allelic variants feasible. Using VAR2CSA-type PfEMP1—a notoriously polymorphic antigen involved in the pathogenesis of placental malaria—as a model, we demonstrate the robustness of the assay and its applicability to analysis of true cross-reactivity of monoclonal VAR2CSA-specific antibodies in naturally exposed individuals. The assay is adaptable to the analysis of other polymorphic antigens, rendering it a powerful tool in studies of immunity to malaria and many other diseases.

## Editor's evaluation

The information included in the manuscript attests to the intended rigor taken by the investigators in providing a high-quality resource material for the use by scientists interested in antibody-mediated responses to pathogens.

## Introduction

Malaria is a serious infectious disease caused by mosquito-transmitted protozoan parasites of the genus *Plasmodium*. At least five species of these parasites can cause disease in humans, but by far the most serious is *Plasmodium falciparum*. This parasite alone was responsible for an estimated 241 million disease episodes and 627,000 deaths in 2020, mainly in sub-Saharan Africa (**WHO, 2021**).

Acquired immunity to *P. falciparum* malaria following natural exposure is mediated mainly by IgG antibodies with specificity for the asexual blood stages of the infection (**Cohen et al., 1961**;

*Sabchareon et al., 1991*). However, acquisition of protection following natural exposure takes years to develop, and complete protection is rarely if ever achieved (reviewed by *Hviid, 2005*). The extensive inter-clonal polymorphism and intra-clonal variation of key parasite antigens appear to be important reasons. Consequently, the identification of conserved and functionally important antibody epitopes in key antigens is a major goal of malaria immunology research. However, it is a goal that is difficult to achieve using conventional approaches such as analysis of immune sera by enzyme-linked immuno-sorbent assay (ELISA). A major obstacle is the inability to separating true (i.e., a single antibody that recognizes a conserved epitope shared by multiple allelic variants; *Figure 1A*) from apparent cross-reactivity (multiple antibodies, each recognizing a variant-specific epitope; *Figure 1B*), because this requires analysis at the single B-cell level.

The development of a B-cell ELISpot assay was a first step toward that (*Czerkinsky et al., 1983*). In the assay, each spot corresponds to the antibodies secreted by a single B-cell, that is, to a monoclonal antibody. The ELISpot assay has since been modified to allow detection of the antibody-secreting B cells (ASCs) by fluorescence and concomitant detection of several antibody specificities (reverse FluoroSpot) (*Hadjilaou et al., 2015*). Further modifications enabled the determination of antibody cross-reactivity against different dengue serotypes and closely related viruses (*Adam et al., 2018*), and a versatile adaptation of the assay to antigens of choice (*Jahnmatz et al., 2016*). In the present report, we describe a further derivation of the FluoroSpot assay that combines these advantages and validates its performance using allelic variants of highly polymorphic VAR2CSA-type PfEMP1 (*Bock-horst et al., 2007*; *Salanti et al., 2003*; *Trimnell et al., 2006*). PfEMP1 is a family of proteins that appears to be an antibody target of central importance to acquisition of clinical immunity to *P. falciparum* malaria (reviewed by *Stevenson et al., 2015*). Despite the undoubted clinical importance of particular types of PfEMP1 in the pathogenesis of severe malaria complications such as cerebral and placental malaria (*Jensen et al., 2004*; *Lennartz et al., 2017*; *Salanti et al., 2004*; *Turner et al., 2013*), the extensive polymorphism of these antigens jeopardizes the development of efficacious PfEMP1-based vaccines (reviewed by *Hviid et al., 2018*). VAR2CSA is responsible for the accumu-lation of infected erythrocytes (IEs) in the placenta (*Salanti et al., 2004*). This can lead to placental malaria, which affects about a third of all pregnancies in malaria-endemic areas and is the direct cause of substantial maternal, fetal, and infant morbidity (WHO, 2021). It was recently reported that vaccines, based on the minimal binding domain that is assumed to be a functionally conserved region of VAR2CSA mediating placental IE sequestration by binding to oncofetal chondroitin sulfate A (CSA), induced a variant-specific rather than cross-reactive antibody response (*Mordmüller et al., 2019*; *Sirima et al., 2020*). These findings underscore the need to identify genuinely cross-reactive antibody epitopes. The new 'plug-and-play' FluoroSpot assay described and validated here facilitates such an effort. While applied here to the study of cross-reactivity with allelic variants of VAR2CSA, it can easily be adapted for examination of other polymorphic antigens that are known or suspected targets of antibody-mediated protective immunity to malaria and other infectious diseases.

## Results
### Production and quality control of recombinant antigens
We produced recombinant proteins corresponding to four allelic variants (IT4, NF54, M920, and Malayan Camp) of the ID1-ID2, DBL3X, and DBL5ε regions of *P. falciparum* VAR2CSA in human embry-onic kidney cells. Each recombinant antigen included one of four peptide tags (GAL, TRAP, TWIN, and WASP) to enable 'plug-and-play' detection with standard, tag-specific, and fluorescently labeled reagents (*Figure 1C*). All the recombinant proteins appeared as single bands at the expected sizes in Instant Blue-stained SDS-PAGE gels (*Figure 1—figure supplement 1*). All the recombinant antigens were specifically recognized in the sex-specific manner (recognized by plasma IgG from *P. falciparum*-exposed women but not from sympatric men) typical of VAR2CSA (*Fried et al., 1998*; *Ricke et al., 2000*; *Salanti et al., 2004*), when tested in ELISA (*Figure 1—figure supplement 2*). The IT4 ID1-ID2 allelic variant was produced with each of the four tags to allow testing of the performance of the assay in various configurations (see below). Each of the differently tagged versions of IT4 ID1-ID2 reacted identically with the different plasma IgG pools (*Figure 1—figure supplement 2A*). The allelic variants of DBL3X and DBL5ε were therefore produced with one tag only (*Figure 1C*). We conclude that the quality of all the VAR2CSA region constructs produced was satisfactory.

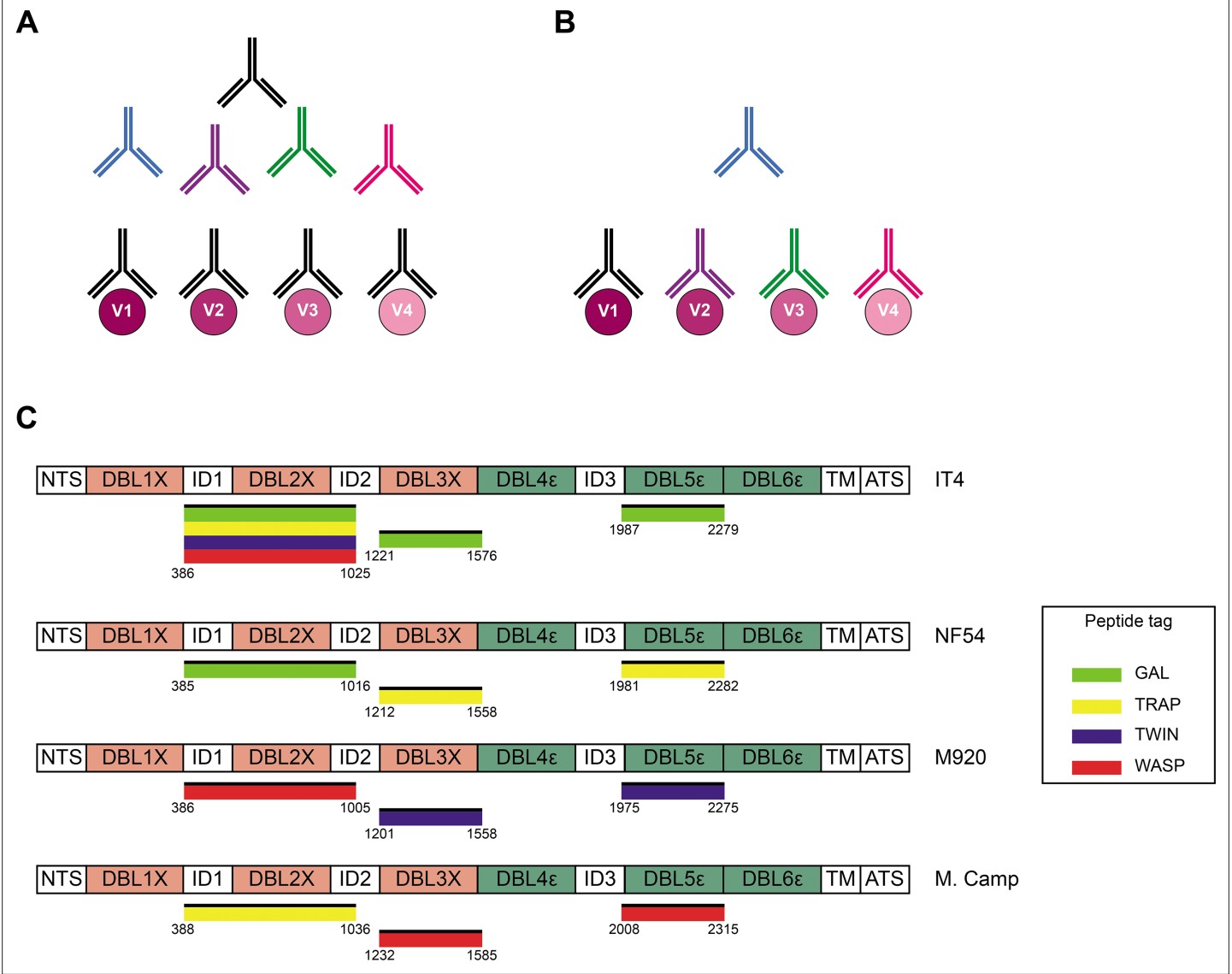

**Figure 1.** Cross-reactivity in the VAR2CSA-specific response present in serum. (**A**) Schematic representing 'true' cross-reactivity where an epitope shared among different VAR2CSA variants (V1–V4) is recognized by a single monoclonal antibody (colored in black) present in the polyclonal serum. (**B**) In 'apparent' cross-reactivity a VAR2CSA polymorphic epitope is recognized by several different variant-specific monoclonal antibodies present in serum. (**C**) Schematic structure of the VAR2CSA variants included in this study, depicting the N-terminal segment (NTS), the six Duffy binding-like (DBL) domains, the interdomain (ID) regions, the transmembrane domain (TM), and the intracellular acidic terminal segment (ATS). The relevant domain boundaries are indicated with amino acid positions under each schematic. The peptide-tagged recombinant domains are also indicated under the sequence with the corresponding fluorophore colors used for detection in the reversed FluoroSpot assay.

The online version of this article includes the following source data and figure supplement(s) for figure 1:

**Figure supplement 1.** Instant blue-stained gels of purified, recombinant peptide-tagged VAR2CSA domains under reducing (+DTT) and non-reducing conditions (−DTT).

**Figure supplement 1—source data 1.** Uncropped images used to build *Figure 1—figure supplement 1*.

**Figure supplement 1—source data 2.** Original picture file for the instant blue-stained gel of the purified recombinant GAL-tagged VAR2CSA domains.

**Figure supplement 1—source data 3.** Original picture file for the instant blue-stained gel of the purified recombinant TRAP-tagged VAR2CSA domains.

**Figure supplement 1—source data 4.** Original picture file for the instant blue-stained gel of the purified recombinant TWIN-tagged VAR2CSA domains.

**Figure supplement 1—source data 5.** Original picture file for the instant blue-stained gel of the purified recombinant WASP-tagged VAR2CSA domains.

*Figure 1 continued*

**Figure supplement 2.** IgG-binding to recombinant VAR2CSA domains by ELISA.

**Figure supplement 3.** Antibody binding to recombinant VAR2CSA domains by ELISA.

**Figure supplement 4.** FluoroSpot detection reagents binding to recombinant VAR2CSA domains by ELISA.

We next used ELISA to test the recognition of the recombinant antigens by monoclonal antibodies with known specificity for the relevant regions of VAR2CSA. The mouse monoclonal antibody 6E2, derived from a mouse immunized with recombinant IT4 ID1-ID2 (unpublished data), reacted with the IT4, NF54, and M920 allelic variants of ID1-ID2, but did not react with Malayan Camp ID1-ID2 (*Figure 1—figure supplement 3A*). The DBL3X-specific human monoclonal antibodies PAM2.8 and PAM8.1 reacted with all four allelic variants of DBL3X, although PAM8.1 recognized NF54 DBL3X is less well than the other variants. The DBL5ε-specific human monoclonal antibody PAM3.10 recognized all four variants of VAR2CSA-DBL5ε equally well. PAM1.4, reported to recognize a conserved, but conformational and probably discontinuous epitope in VAR2SCA (*Barfod et al., 2007*), did not recognize any of the VAR2CSA region constructs (*Figure 1—figure supplement 3B*). All the tested antibodies, including PAM1.4, recognized a recombinant protein representing the full ectodomain of IT4 VAR2CSA (also known as IT4VAR04). A DBL4ε domain from a non-VAR2CSA PfEMP1 protein was not recognized by any of the monoclonal antibodies, underscoring their VAR2CSA-specificity. Taken together, these findings correspond fully with previous evidence regarding the allelic variant-specificity of the monoclonal antibodies (*Barfod et al., 2010*), and thus further underpin the quality of the peptide-tagged antigens produced here.

As a final step in the quality assessment of the peptide-tagged antigens, we assessed the ability of the tag detection reagents to identify them by ELISA (*Figure 1—figure supplement 4*). Most of the tagged proteins were recognized exclusively by the detection reagent corresponding to the incorporated tag. However, the TRAP-specific detection reagent showed minor cross-reactivity with the WASP-tagged Malayan Camp DBL3X antigen and the M920 DBL5ε allelic variant was not well recognized. We conclude from these experiments that the tagging of the antigens was adequate, except for M920 DBL5ε.

## Quality control of assay performance

We first analyzed the performance of the FluoroSpot assay without any multiplexing, that is, employing a single antigen and a single detection reagent. This configuration (1×1) corresponds to a basic reverse ELISpot assay, except for the use of a fluorescent rather than an enzymatic detection system. For this analysis, we used a mouse hybridoma secreting the ID1-ID2-specific monoclonal antibody 6E2, a tagged IT4 ID1-ID2 antigen and the corresponding tag-specific detection reagent. For each matching tag/detection reagent combination (*Figure 2—figure supplement 1A*), we detected similar numbers of spots, each corresponding to a 6E2-secreting hybridoma cell (*Figure 2A*). Spots were exclusively detected in the instrument channel corresponding to the detection reagent used (*Figure 2—figure supplement 2A*). In other words, only single-colored spots of the expected color were detected. These results documented the assay's ability to detect individual ASC by the reactivity of the secreted monoclonal antibody with cognate tagged antigen. Detection was highly specific and independent of the tag/detection reagent combination used.

We next tested 1×4 (one tagged antigen assayed with four different detection reagents) and 4×1 (four differently tagged versions of one antigen assayed with one detection reagent) configurations of the assay (*Figure 2—figure supplement 1B-C*). For these experiments, we used the 6E2-secreting hybridoma, tagged IT4 ID1-ID2 antigens, and detection reagents as above. In each of the 1×4 configurations (*Figure 2—figure supplement 1B*), spots were exclusively detected in the correct channel (*Figure 2—figure supplement 2B*), with similar spot counts among the different 1×4 designs and corresponding to the results obtained with the 1×1 design (*Figure 2A*). The results document that the presence of multiple detection reagents did not affect assay performance. In each of the four complementary (4×1) configurations (*Figure 2—figure supplement 1C*), again spots were exclusively detected in the correct channel (*Figure 2—figure supplement 2C*), with similar spot counts regardless of the detection reagent used, and corresponding to the results obtained with the 1×1 and

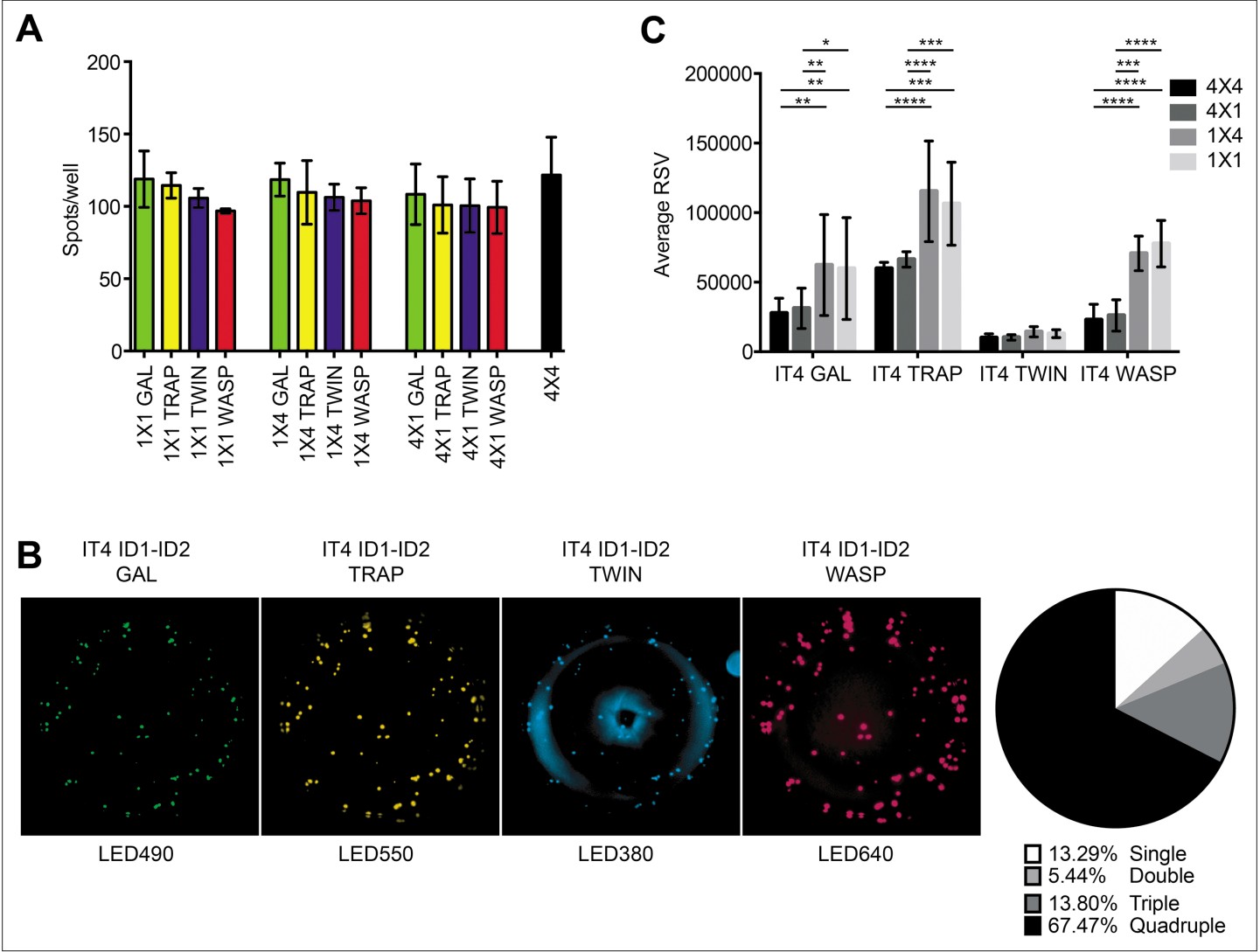

**Figure 2.** FluoroSpot analysis of the 6E2 monoclonal antibody against the IT4 ID1-ID2 domain. Captured antibodies secreted by 6E2 cells were incubated with peptide-tagged IT4 ID1-ID2 domains and detected using peptide-specific detection reagents. (**A**) Different antigen×detection reagent combinations were tested (1×1, 1×4, 4×1, and 4×4) and the number of spots/well were counted. (**B**) Images acquired in each detection channel corresponding to each detection reagent are presented (LED490, LED550, LED380, and LED640) for the spots detected in the 4×4 combination (black bar in (**A**)). Percentages of single, double, triple, and quadruple positive spots are also presented. (**C**) Average relative spot volume (RSV) for all the positive spots per antigen was determined and compared between the assay combinations. Means and standard deviations of data from three independent experiments are shown.

The online version of this article includes the following figure supplement(s) for figure 2:

**Figure supplement 1.** Schematic showing the different antigen×detection reagent combinations tested, corresponding to the data presented in *Figure 2*.

**Figure supplement 2.** Well images for the different antigen×detection reagent combinations tested, corresponding to the schematic presented in *Figure 2—figure supplement 1* and the data presented in *Figure 2*.

1×4 configurations (*Figure 2A*). The results document that the simultaneous use of multiple tagged versions of the same antigen did not affect assay performance.

Finally, we tested the assay in its intended 4×4 configuration, using four differently tagged versions of the same antigen, assayed with four different detection reagents (*Figure 2—figure supplement 1D*). Cells and reagents for these experiments were the same as above. As in the previous configurations, we detected similar numbers of spots (*Figure 2B*), corresponding closely to those detected in the 1×1, 1×4, and 4×1 configurations. Of note, most of the spots were detected by at least two (and

primarily by four) of the tag detection reagents (*Figure 2B*). The analysis software used also allowed analysis of the relative spot volume (RSV), which is a calculated value integrating information about the size and fluorescence intensity of each spot. The RSV thus reflects the amount and affinity of the secreted monoclonal antibody (*Jahnmatz et al., 2020a*). Although spot numbers were consistent in all configurations, RSV values were consistently higher for configurations with only one tagged antigen (1×1 and 1×4) than configurations with differently tagged versions of the antigen (4×1 and 4×4) (*Figure 2C*). This likely reflects competition among the differently tagged versions of the IT4 ID1-ID2 antigen for binding to captured 6E2 antibodies when all versions were added together. We consistently observed an apparent higher level of background signal in the LED380 detection channel, evidenced by a ring of blue fluorescence at the edge and the center of the wells that occasionally generated an increase in the number of single-positive spots in this channel (likely false positives). This observation suggests that this channel could be left out if a higher accuracy is desired, reducing the number of antigen variants tested to three. However, this would obviously reduce the ability of the assay to assess broad cross-reactivity, particularly when highly diverse antigens are being studied. We conclude from these experiments that our FluoroSpot assay produced reliable and consistent results, regardless of the complexity of multiplexing.

## Validation of assay performance

Our goal with the FluoroSpot assay validated above was to use it for the determination of the degree of 'true' (monoclonal) antibody cross-reactivity with allelic variants of the same antigen. We tested this application with four allelic variants (IT4, NF54, M920, and Malayan Camp) of three regions (ID1-ID2, DBL3X, and DBL5ε) of VAR2CSA. When all four allelic variants of ID1-ID2 and all four detection reagents were added to secreted and captured 6E2 antibody (*Figure 3—figure supplement 1A*), spots were only detected with the IT4 allele reagent (TWIN) and M920 allele reagent (WASP) (*Figure 3A*), indicating the presence of a strongly 6E2-crossreactive epitope present in IT4 and M920, which is absent from NF54 and Malayan Camp. This conclusion was supported by the fact that most of the detected spots were double-positive. However, in the initial ELISA experiments, 6E2 reacted with the NF54 allele of ID1-ID2, in addition to its reaction with the IT4 and M920 alleles (*Figure 1—figure supplement 3*). We therefore tested if the absence of 6E2 reactivity to NF54 ID1-ID2 in the FluoroSpot assay was related to the multiplexing, despite the assay validation experiments described above. The results obtained with 1×1, 1×4, and 4×1 configurations (*Figure 3—figure supplement 1B-D*), corresponded very well with the results obtained with the 4×4 configuration (*Figure 3A*), although we did detect a few faint spots with the NF54 ID1-ID2-specific detection tag (GAL) in the 4×1 configuration (*Figure 3B*). In agreement with the only partial cross-reactivity of 6E2 with the different allelic variants of ID1-ID2, RSV values were less affected by multiplexing (*Figure 3C*) than when assaying 6E2 with differently tagged versions of the same ID1-ID2 allele (IT4) (*Figure 2C*), consistent with a lower degree of antigen competition for bound antibody. These results show that detection of true allelic cross-reactivity of monoclonal antibodies is possible with the FluoroSpot assay, although some low-affinity cross-reactivity might be missed.

We next performed experiments with four human B-cell clones derived from Epstein-Barr virus immortalized memory B cells obtained from women with natural exposure to placental malaria. These experiments were done to substantiate the versatility of the assay, in particular its application to analysis of human B cells and antibodies. The clones PAM2.8 and PAM8.1 react with several allelic variants of DBL3X, while PAM 3.10 reacts with several DBL5ε variants (*Barfod et al., 2007*; *Barfod et al., 2010*). The last clone, PAM1.4, reacts with most allelic variants of full-length VAR2CSA but does not react with any single-domain DBL constructs from any VAR2CSA variant tested so far. It was therefore included here as a negative control.

In the fully multiplexed 4×4 configuration of the assay (*Figure 4—figure supplement 1A*), the PAM2.8-secreting clone produced spots that could be detected with all four allelic variants of DBL3X, with most spots reacting with three or four of the detection reagents (*Figure 4A*). The PAM8.1 antibody reacted with the IT4, M920, and Malayan Camp allelic variants of DBL3X, but was non-reactive with NF54 DBL3X as expected, since this particular variant lacks the linear epitope recognized by PAM8.1. Most spots reacted with two or three detection reagents (*Figure 4B*). In the same assay configuration, but employing allelic variants of DBL5ε rather than DBL3X (*Figure 4—figure supplement 1A*), PAM3.10 was shown to cross-react with all four allelic variants DBL5ε, and mainly produced

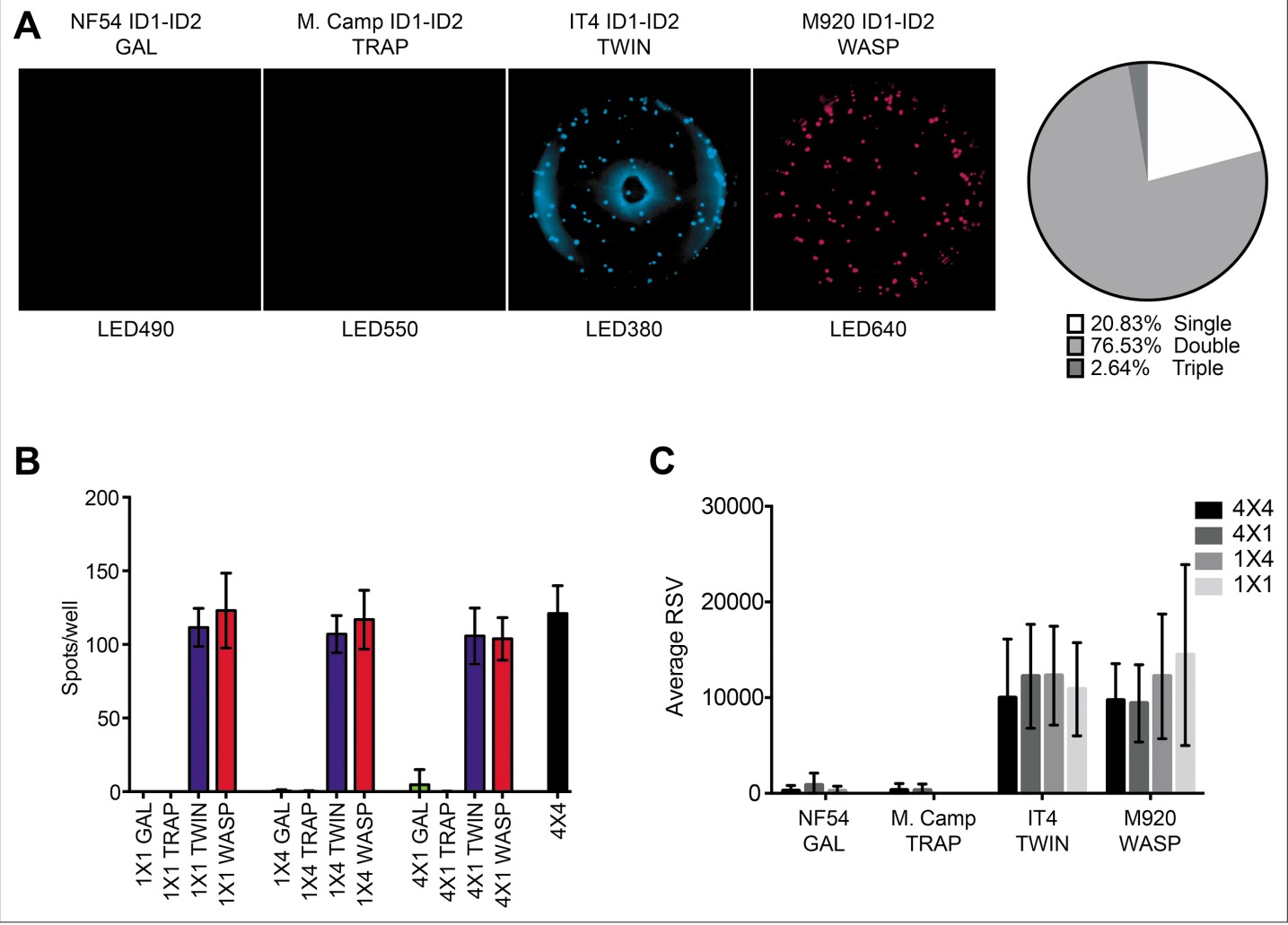

**Figure 3.** FluoroSpot analysis of the 6E2 monoclonal antibody against four peptide-tagged ID1-ID2 domain variants (NF54, M.Camp, IT4, and M920). (**A**) Captured 6E2 antibodies secreted by 6E2 cells were incubated with all four peptide-tagged ID1-ID2 domain variants and detected using the four peptide-specific detection reagents (multiplex 4×4). Images acquired in each detection channel are presented (LED490, LED550, LED380, and LED640) together with the percentages of single, double, triple, and quadruple positive spots. (**B**) Different antigen ×detection reagent combinations were tested (1×1, 1×4, 4×1, and 4×4) and the number of spots/well were counted. (**C**) Average relative spot volume (RSV) for all the positive spots per antigen was determined and compared between the assay combinations. Means and standard deviations of data from three independent experiments are shown.

The online version of this article includes the following figure supplement(s) for figure 3:

**Figure supplement 1.** Schematic showing the different antigen×detection reagent combinations tested, corresponding to the data presented in *Figure 3*.

**Figure supplement 2.** Well images for the different antigen×detection reagent combinations tested, corresponding to the schematic presented in *Figure 3—figure supplement 1* and the data presented in *Figure 3*.

spots reacting with three or four of the detection reagents (*Figure 4C*). Results of supplementary experiments with the more restricted configurations of the assay (*Figure 4—figure supplement 1B-D*) were in broad agreement with the 4×4 configuration results (*Figure 4—figure supplements 2–4*). We did not observe any spots when testing the PAM1.4-secreting clone in the FluoroSpot assay (4×4 configuration) with any of our DBL3X or DBL5ε allelic variants (data not shown). All these results were consistent with those obtained with ELISA (*Figure 1—figure supplement 3*) and with previous reports on the specificity of these monoclonal antibodies (*Barfod et al., 2007*; *Barfod et al., 2010*). RSV values were consistently higher for configurations with only a single tagged allelic variant (1×1 and 1×4) than for configurations with multiple, differently tagged allelic variants of the antigen present

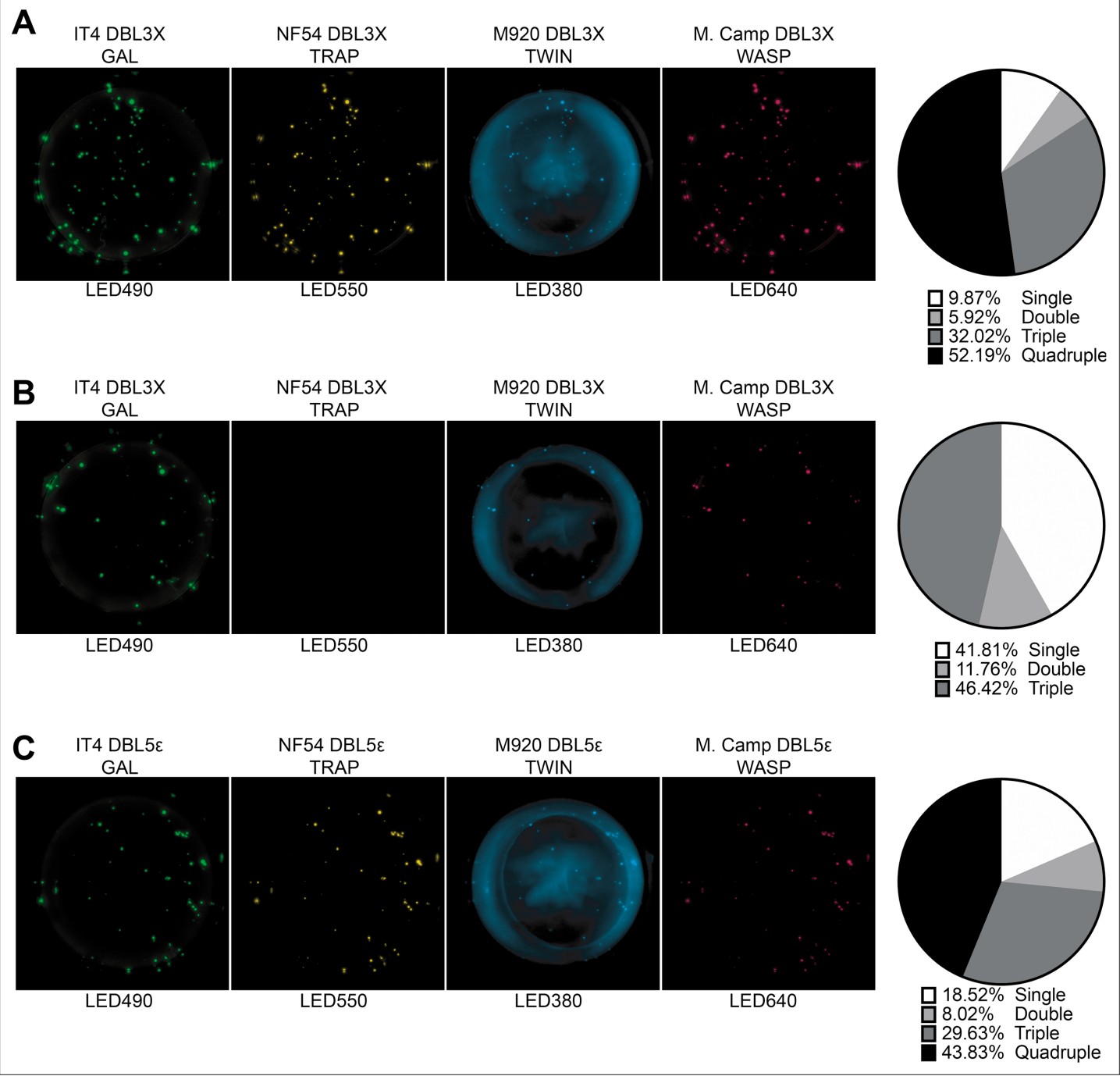

**Figure 4.** FluoroSpot analysis of the VAR2CSA specific antibodies PAM2.8, PAM8.1, and PAM3.10 against four peptide-tagged DBL3X and DBL5ε domain variants (IT4, NF54, M920, and M.Camp). Captured antibodies secreted by the corresponding EBV-immortalized cell lines were incubated with all four peptide-tagged DBL3X (PAM2.8 and PAM8.1) or DBL5ε (PAM3.10) domain variants and detected using the four peptide-specific detection reagents (multiplex 4×4). (**A**) PAM2.8, (**B**) PAM8.1, and (**C**) PAM3.10. Images acquired in each detection channel are presented (LED490, LED550, LED380, and LED640) together with the percentages of single, double, triple, and quadruple positive spots.

The online version of this article includes the following figure supplement(s) for figure 4:

**Figure supplement 1.** Schematic showing the different antigen×detection reagent combinations tested (**A**) 4×4 (corresponding to the data presented in *Figure 4*), (**B**) 1×1, (**C**) 1×4, and (**D**) 4×1.

**Figure supplement 2.** Well images for the different antigen×detection reagent combinations tested for PAM2.8, corresponding to the schematic presented in *Figure 4—figure supplement 1*.

*Figure 4 continued on next page*

*Figure 4 continued*

**Figure supplement 3.** Well images for the different antigen×detection reagent combinations tested for PAM8.1, corresponding to the schematic presented in *Figure 4—figure supplement 1*.

**Figure supplement 4.** Well images for the different antigen×detection reagent combinations tested for PAM3.10, corresponding to the schematic presented in *Figure 4—figure supplement 1*.

**Figure supplement 5.** Comparison of the average relative spot volume (RSV) for all the positive spots per antigen between the assay combinations for (**A**) PAM 2.8, (**B**) PAM8.1, and (**C**) PAM3.10.

---

(4×1 and 4×4) (*Figure 4—figure supplement 5*). On the above basis, we concluded that our assay was able to reliably quantify cross-reactivity of monoclonal antibodies with up to four allelic variants of the same antigen.

## Pilot application of assay

In the final set of experiments, we applied the validated assay to the analysis of samples of peripheral blood mononuclear cells (PBMCs) from seven women with previous natural exposure to VAR2CSA-type PfEMP1 proteins during *P. falciparum* infection-exposed pregnancies (mean parity: 4, range: 2–8). PBMCs from three sympatric men were included as negative controls. After induction of memory B-cell differentiation to ASC as described elsewhere (*Jahnmatz et al., 2013*), the PBMC samples were assayed for ASC secreting IgG of any specificity (total IgG) and IgG antibodies specific for the four allelic variants of the VAR2CSA ID1-ID2, DBL3X, and DBL5ε regions described above. The frequencies of ASC secreting VAR2CSA-specific IgG tended to be higher among women than among the men, where only low frequencies were detected (*Figure 5*). Although the difference was statistically significant for the ID1-ID2 region only (probably due to the low number of individuals tested), the pattern was the same for all the VAR2CSA regions tested and fully consistent with the well-established pregnancy-dependency of acquisition of substantial antibody reactivity to VAR2CSA. Most of the spots detected were single-colored (*Figure 5* and *Figure 5—figure supplement 1*), indicating that most corresponded to antibodies recognizing variant-specific epitopes not conserved among the tested allelic variants. For five of the samples, sufficient cells were available to also test them in the 1×1 configuration. No systematic differences in spot numbers or RSV values were detected between the 4×4 and 1×1 configurations (*Figure 5—figure supplement 2*). We conclude that the assay performed as expected, but that larger studies will be required to draw detailed conclusions regarding the level of antibody cross-reactivity with allelic variants of polymorphic antigens present in samples collected from naturally exposed women.

## Discussion

Clinical protection against malaria in areas with stable transmission of *P. falciparum* parasites is acquired in a piecemeal fashion over several years. This slow rate of acquisition, and the fact that sterile protection following natural exposure is very rarely achieved, is often attributed to the very substantial polymorphic (allelic or inter-clonal) variation of the antigens that are important for protection (reviewed by *Hviid, 2005*). It is assumed that allelic variation represents an important immune-evasive strategy of the malaria parasites and that acquisition of broadly cross-reactive antibodies to key antigens following natural exposure is rare. Instead, acquisition of protection is thought to rely on the accumulation of a broad repertoire of antibodies, which may each be variant-specific, but which together cover the repertoire of allelic antigen variants (*Bull et al., 1998*; *Nielsen et al., 2002*).

The above scenario has prompted a search for epitopes that are shared among multiple allelic variants (i.e., conserved epitopes) in key antigens. Identification of conserved epitopes is of particular importance in the development of efficacious vaccines against malaria, not least those based on PfEMP1. Although these vaccine candidates have the advantage of being based on antigens with well-documented roles in malaria pathogenesis and acquired immunity to various severe manifestation of the disease, they suffer from the presence of numerous allelic variants, thought by many to preclude their utility in vaccine development (reviewed by *Hviid et al., 2018*). The recent reports documenting marked variant-specificity of antibodies induced by vaccination with the supposedly conserved minimal binding domain of VAR2CSA illustrates this concern (*Mordmüller et al., 2019*; *Sirima et al., 2020*). This notwithstanding, it is evident that VAR2CSA-specific B cells that secrete

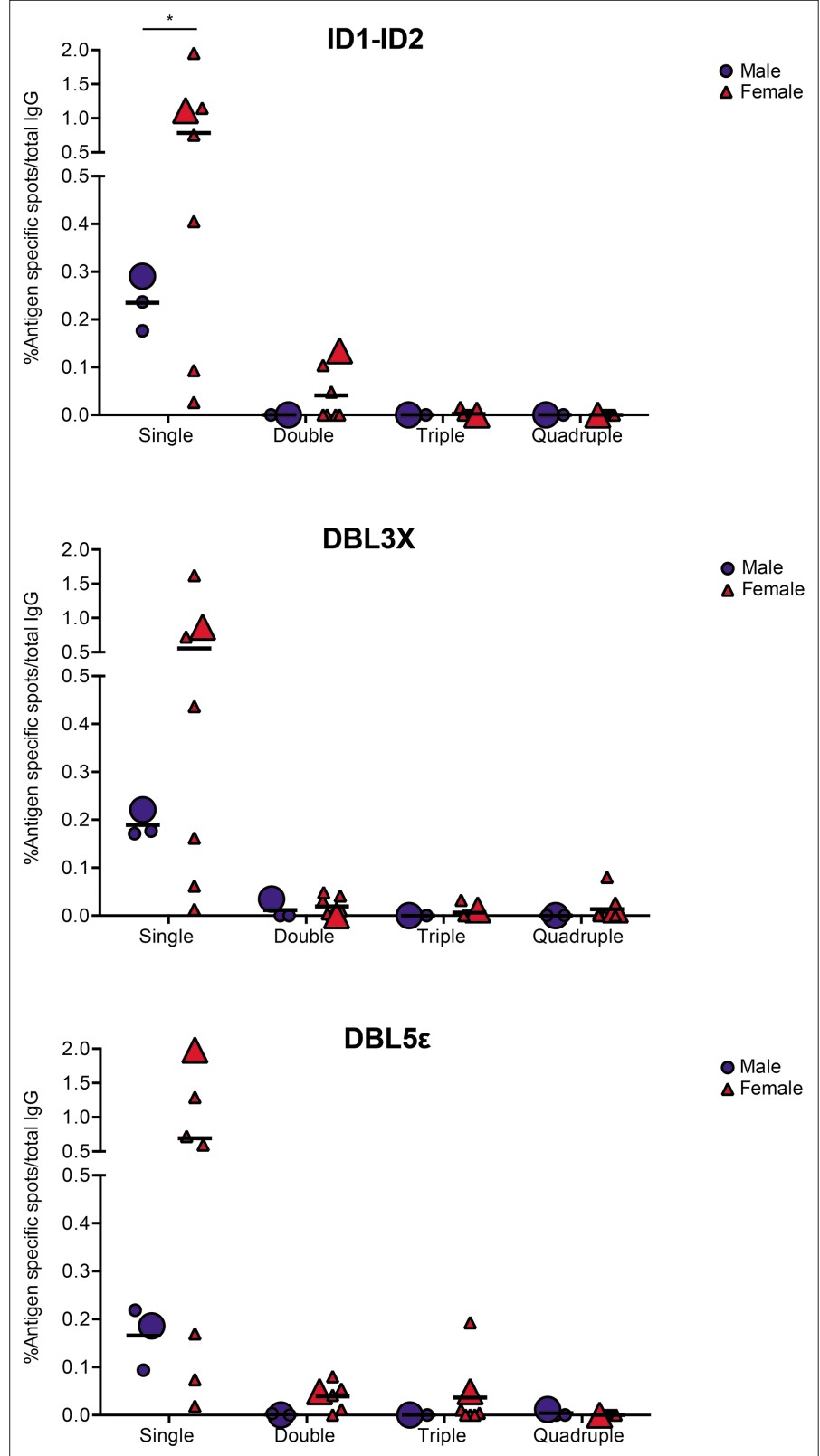

**Figure 5.** VAR2CSA specific responses in malaria-exposed Ghanaian donors against four ID1-ID2 (**A**), DBL3X (**B**), and DBL5ε (**C**) variants tested simultaneously. The scatter dot plots depict the percentages of antigen specific spots/total IgG secreting cells for each individual donor tested discriminating variant specific (single-coloured) and

*Figure 5 continued on next page*

*Figure 5 continued*

cross-reactive (double-, triple-, and quadruple-positive) spots. Female and male donors are compared highlighting (with larger symbols) one representative donor for each group.

The online version of this article includes the following figure supplement(s) for figure 5:

**Figure supplement 1.** Images acquired in each detection channel are presented (LED490, LED550, LED380, and LED640) for the two representative donors highlighted in *Figure 5*.

**Figure supplement 2.** Comparison of number spots/well (**A**) and average RSV (**B**) for each of the four VAR2CSA variants and three domains tested, either in a 4×4 (circles) or a 1×1 (triangles) configuration.

broadly cross-reactive antibodies do exist. Thus, memory B cells from women with natural exposure to placental malaria can be induced to secrete truly cross-reactive human monoclonal antibodies that are specific for VAR2CSA-type PfEMP1—in fact, all of the monoclonal antibodies described showed substantial allelic variant cross-reactivity (*Barfod et al., 2007*; *Barfod et al., 2010*). VAR2CSA thus contains conserved epitopes.

On the above basis, we set out to develop and test an assay that would facilitate in-depth analysis of antibody cross-reactivity with allelic variants of the same polymorphic antigen. We chose VAR2CSA-type PfEMP1 as our model antigen, and included four allelic variants, obtained from the strains IT4, NF54, M920, and Malayan Camp. These strains were chosen because they are geographically and temporally distant, and because their amino acid sequences belong to distinct phylogenetic clusters (*Renn et al., 2021*). As all PfEMP1 antigens, VAR2CSA contains several Duffy binding-like (DBL) domains, some separated by so-called interdomain (ID) regions (*Stevenson et al., 2015*; *Figure 1C*). For each of the chosen VAR2CSA variants, we expressed three regions: DBL3X, DBL5ε, and the ID1-ID2 region (*Figure 1C*). DBL3X and DBL5ε were chosen as they appear to be dominant targets of naturally acquired and truly cross-reactive VAR2CSA-specific IgG (*Barfod et al., 2010*), while the ID1-ID2 region was selected because it includes the above-mentioned minimal binding region used as antigen in the recent vaccine trials (*Mordmüller et al., 2019*; *Sirima et al., 2020*). It is defined as the shortest contiguous part of VAR2CSA that retains the affinity of the full-length molecule for the VAR2CSA cognate receptor, oncofetal CSA (*Clausen et al., 2012*). The choice of DBL3X, DBL5ε, and ID1-ID2 was furthermore influenced by the convenient availability of cell lines producing monoclonal antibodies specific for them.

The multiplexed reverse FluoroSpot assay has previously been applied to studying the reactivity of monoclonal antibodies against four serotypes of dengue virus (DENV-1 to DENV-4) (*Adam et al., 2018*; *Hadjilaou et al., 2015*). One study detected the binding of DENV-1 to DENV-4 virions to immobilized monoclonal antibody via a panel of DENV serotype-specific monoclonal antibodies labeled by different fluorochromes (*Hadjilaou et al., 2015*). The other study instead used fluorescently labeled virus particles representing the four dengue serotypes. We have previously described a similar assay that is easier to adapt to any antigen combination of choice than the above approaches, because it employs recombinantly tagged antigens with discrete peptides that can be detected by standardized tag-specific reagents (*Jahnmatz et al., 2016*). The assay is thus more 'Plug-and-Play', as tagged antigens are fairly easy to generate. That version of the assay has been successfully applied to simultaneous detection of B cells secreting antibodies specific for differently tagged antigens from three different pathogens (*Jahnmatz et al., 2020a*) and for four distinct *P. falciparum* antigens (*Jahnmatz et al., 2020b*). The assay has not previously been used for the analysis of antibody cross-reactivity to allelic variants of a single antigen, which is more demanding in terms of specificity (ability to distinguish between allelic variants rather than different antigens altogether). Through a series of experiments documenting the performance of the assay in multiple configurations, we could demonstrate its applicability to studying true (monoclonal) antibody allelic cross-reactivity. We furthermore documented the adaptability of the assay by applying it to the study of allelic variants of three different regions (ID1-ID2, DBL3X, and DBL5ε) of VAR2CSA. Finally, we applied the assay to a pilot analysis of antibody allelic cross-reactivity among B cells from individuals with natural exposure to *P. falciparum*. Most of the B cells that secreted antibodies recognizing our test antigens only recognized a single allelic variant. This finding is in agreement with a previous study (*Doritchamou et al., 2016*). The authors of the earlier study sequentially depleted immune sera of reactivity with allelic variants of ID1-ID2 and DBL5ε (and of DBL4ε, not included in our study). They concluded, like we do here, using a fundamentally different approach, that allelic cross-reactivity of VAR2CSA-specific IgG appears to

be very limited. This would explain the very limited cross-reactivity observed in the recent VAR2CSA vaccine trials (*Mordmüller et al., 2019*; *Sirima et al., 2020*).

While the current study was in progress, *Doritchamou et al., 2022* reported a new study, using the same approach as in their earlier study (*Doritchamou et al., 2016*), but now employing allelic variants of the full ectodomain of VAR2CSA (*Renn et al., 2021*) rather than the single- and oligo-domain antigens used previously (and here). The new study indicated a very marked cross-reactivity of VAR2CSA-specific antibodies, as depletion on a single allelic variant resulted in almost complete loss of reactivity with the other variants. This finding is in stark contrast to those reported here as well as in the authors' own earlier study. However, it agrees very well with our earlier reports (*Barfod et al., 2007*; *Barfod et al., 2010*) that VAR2CSA-specific memory B cells generated in response to natural parasite exposure tend to secrete antibodies that are broadly cross-reactive.

Taken together, these results suggest that broadly cross-reactive antibodies to VAR2CSA and probably to other PfEMP1 antigens—indeed, possibly to many complex, high-molecular weight proteins—target conformational epitopes that are not properly reproduced by smaller antigens consisting of domains or regions of the larger, full-length protein. Some of these—such as the epitope of the monoclonal antibody PAM1.4, which recognizes the full ectodomain of most VAR2CSA variants but not any of its constituent domains individually (*Barfod et al., 2007*)—are likely to be composed of amino acids that are far apart in the primary sequence of the protein. However, there may be more to it than that. The PAM2.8, PAM3.10, and PAM8.1 antibodies were all obtained from B cells of individuals exposed to—and were identified by screening for reactivity with—IEs displaying native (i.e., full-length) VAR2CSA (as were PAM1.4). It is an open question whether those antibodies—even though they were subsequently shown to react only with epitopes in either DBL3X (PAM2.8 and PAM8.1) or DBL5ε (PAM3.10), would have been identified by screening for reactivity with these single domain antigens. The high degree of variant-specificity observed here and by *Doritchamou et al., 2016* suggests that the answer is no. It also suggests that what determines the fine specificity of antibodies elicited after exposure to native complex antigens may not be the same as that determining fine specificity after subunit vaccination, even when those antibodies all recognize epitopes within the construct used for vaccination.

In conclusion, we have developed an assay that is suitable for the interrogation of the degree of allelic cross-reactivity of monoclonal antibodies. However, our results indicate that it may also be applied to deeper investigations of how antibody fine specificity (including variant specificity versus cross-reactivity) is determined. The main caveat with respect to the conclusions drawn here is that our work was mainly a methods development study and therefore involved only a relatively small number of individuals. Larger studies should be undertaken to assess the robustness of our preliminary conclusions regarding the frequency of allelic cross-reactivity in pregnant women naturally exposed to *P. falciparum* infection.

## Materials and methods

### Recombinant antigens

Plasmids encoding the full-length VAR2CSA ectodomains of *P. falciparum* strains FCR3/IT4, 3D7/NF54, M920, and Malayan Camp, codon-optimized for expression in mammalian cell lines, were used (*Renn et al., 2021*). The DBL3X, DBL5ε, and ID1-ID2 domain sequences were amplified from these plasmids (*Figure 1—figure supplement 1*), using specific primers containing *AflIII* and *XhoI* restriction sites (*Supplementary file 1*).

The amplified sequences were cloned into modified pcDNA3.1 expression vectors. The vectors contained a Kozak consensus sequence, followed by a start codon and a sequence encoding a human serum albumin signal peptide (MKWVTFISLLFLFSSAYSLK). These elements were followed by a multiple cloning site, including the restriction sites *AflII* and *XhoI*, and an SG4S linker to separate the detection tag from the cloned sequence. The four C-terminal detection tags used were GAL (CYPGQ-APPGAYPGQAPPGA), TRAP (DDFLSQQPERPRDVKLA), TWIN-Strep (WSHPQFEKGGGSGGGSGGSA WSHPQFEK), and WASP (CPDYRPYDWASPDYRD) (*Jahnmatz et al., 2016*; *Jahnmatz et al., 2020b*), each followed by a stop codon. The IT4 ID1-ID2 antigen was produced in four versions, tagged with each of the four detection tags (*Figure 1C*). The remaining three ID1-ID2 variants (NF54, M920, and

Malayan Camp) and each of the four DBL3X and DBL5ε domains were each produced with a single tag only (*Figure 1C*).

The generated plasmids were transfected into FreeStyle 293F cells (Thermo Fisher Scientific) using the FreeStyle MAX reagent. Culture supernatants were harvested by centrifugation (3500×*g*, 15 min), 5 days after transfection. ID1-ID2-containing supernatants were buffer exchanged (20 mM HEPES, pH 7.4, 1 mM EDTA, 5% glucose) and the proteins purified by affinity chromatography using a HiTrap Heparin High-Performance column (GE Healthcare) and gradient elution (0–100% NaCl) with 20 mM HEPES pH 7.4, 1 mM EDTA, 5% Glucose, and 1 M NaCl. Selected fractions containing ID1-ID2 were collected, and buffer exchanged into 20 mM HEPES pH 6.5, 1 mM EDTA, and 5% Glucose for further purification by cation-exchange chromatography using HiTrap SP High-Performance columns (GE Healthcare), using a gradient elution with 20 mM HEPES pH 6.5, 1 mM EDTA, 5% Glucose, and 1 M NaCl as described above. The DBL3X and DBL5ε-containing supernatants were buffer exchanged into 20 mM HEPES pH 6.5, 1 mM EDTA, 5% Glucose and the proteins purified by cation-exchange chromatography using a HiTrap SP High-Performance columns (GE Healthcare), followed by size exclusion chromatography using a HiLoad 16/600 Superdex 75 pg column (GE Healthcare). The purified proteins were analyzed by sodium dodecyl sulfate-polyacrylamide gel electrophoresis (SDS-PAGE) under reducing and non-reducing conditions (±DTT) followed by InstantBlue (Expedeon) staining.

Full-length FCR3/IT4 VAR2CSA (IT4 FL), IT4VAR09 (Non-PM IT4 FL), and the single domain PF07_0139 DBLε4 (Non-PM DBLε4) were used as controls for some ELISAs and were expressed and purified as previously described (*Larsen et al., 2018*; *Quintana et al., 2019*; *Stevenson et al., 2015*).

## VAR2CSA-specific monoclonal antibodies

The 6E2 monoclonal antibody is secreted by a hybridoma line derived from a mouse immunized with the ID1-ID2 region of the IT4 variant of VAR2CSA (unpublished data).

The human VAR2CSA-specific monoclonal antibodies PAM1.4, PAM2.8, PAM3.10, and PAM8.1 are secreted by Epstein-Barr virus immortalized memory B cells from PM-exposed Ghanaian women (*Barfod et al., 2007*). PAM1.4 appears to recognize a conserved conformational and discontinuous epitope in VAR2CSA, as it reacts with the majority of native and full-length recombinant VAR2CSA variants, whereas it does not react with individual domains in VAR2CSA. PAM2.8 and PAM8.1 are reported to specifically recognizing epitopes in the DBL3X domain of some, but not all VAR2CSA variants. PAM3.10 recognizes a linear epitope (GKNEKKCINSKS) present in the DBL5ε domain of some, but not all VAR2CSA variants.

## Detection reagents

The following reagents were used to detect the peptide-tags on the recombinant VAR2CSA antigens: rat anti-GAL monoclonal antibody (mAb), mouse anti-TRAP mAb, StrepTactin (to detect the TWIN tag), and mouse anti-WASP mAb. For the FluoroSpot assay, the same but fluorophore-conjugated reagents were used: anti-GAL-490 mAb, mouse anti-TRAP-550 mAb, StrepTactin-380, and mouse anti-WASP-640 mAb. Mabtech AB, Sweden produced all the monoclonal antibodies. The StrepTactin was purchased from IBA-LifeSciences, Germany either as an unconjugated product (to then couple the 380 fluorophore) or as an HRP-conjugate to be used in ELISA.

## Enzyme-linked immunosorbent assays

The specificity of the detection reagents and of the VAR2CSA-specific antibodies listed above against the peptide-tagged VAR2CSA antigens was assessed by ELISA. Briefly, flat-bottomed Nunc MaxiSorp high protein-binding capacity 96-well plates (Thermo Fisher Scientific) were coated overnight at 4°C with the recombinant antigens (0.1 µg/well) prepared in phosphate-buffered saline (PBS). Coating solution was then removed, and each well was blocked with dilution buffer (PBS, 0.5 M NaCl, 1% Triton X-100, 1% bovine serum albumin (BSA), and 0.02 mM phenol red) for 1 hr at room temperature. After three washes (PBS, 0.5 M NaCl, 1% Triton X-100), 50 µL of primary antibody solution (at 10 µg/mL) or dilution buffer were added to each well and incubated for 1 hr at room temperature. After three washes, the plates were incubated with either anti-human-HRP (P214, Dako, 1:6000), anti-rat-HRP (62-9520, Invitrogen, 1:3000), anti-mouse-HRP (P260, Dako, 1:3000) secondary Abs, or StrepTactin-HRP (2-1502-001, IBA-LifeSciences, Germany, 1:10,000), depending on the origin of the primary Ab, for 1 hr at room temperature. Plates were then washed three times and developed by

addition of 3,3′,5,5′-tetramethylbenzidine/TMB PLUS2 (Kementec, Denmark) for 5 min. The reaction was then stopped adding 0.2 M $H_2SO_4$ solution followed by optical density (OD) measurement at 450 nm on a HiPo MPP-96 microplate photometer (BioSan Medical-Biological Research and Technologies, Latvia).

## Cell culture and PBMC samples

Frozen aliquots of the cell lines secreting the VAR2CSA-specific antibodies described above were thawed, washed, and grown in culture media (RPMI 1640, 10% heat-inactivated FBS, 100 U/ml penicillin, 100 µg/ml streptomycin, and 4 mM L-glutamine). Hybridomas and EBV-immortalized human B-cell lines were seeded in T25 flasks at cell densities between 50,000–500,000 and 250,000–1,000,000 cells/mL, respectively, and cultured at 37°C and 5% $CO_2$ until used in the FluoroSpot assay when viability (>90%) and cell densities were counted and adjusted using a hemocytometer after staining with trypan blue. The B cell lines used were previously established in our laboratory by EBV-immortalization followed by limiting dilution to obtain monoclonal lines (*Barfod et al., 2007*). The cells were grown from the master stocks for up to 15–20 passages and were routinely tested (monthly) for the presence of mycoplasma while in use.

Ficoll density centrifugation was used to isolate PBMC from blood samples previously collected from individuals living in a malaria-endemic area (Ghana). Five days before the FluoroSpot assay was performed, PBMC were thawed and cultured at 37°C and 5% $CO_2$ in the presence of 1 µg/mL R848 and 10 ng/mL of recombinant interleukin (IL)2 (both from Mabtech).

## Reverse FluoroSpot assay

The FluoroSpot assay was performed as previously described (*Jahnmatz et al., 2016*; *Jahnmatz et al., 2020a*; *Jahnmatz et al., 2020b*) with small modifications. Low fluorescent 96-well polyvinylidene fluoride sterile plates (Millipore, Bedford, MA) were pre-wetted with 20 µL/well of freshly prepared 40% ethanol solution for 1 min. Ethanol was then thoroughly washed (five times) with 200 µL/well $H_2O$ and one time with PBS. Next, 100 µL of either goat anti-mouse IgG (for 6E2 hybridoma cells) or mouse anti-human IgG (for EBV-immortalized human B cells) specific for the Fc region (3825-3-250 and 3850-3-250, Mabtech) diluted in sterile PBS at 15 µg/mL were added to wells followed by incubation at 4°C for 24 hr. The next day, antibody solution was removed, and wells were washed five times with 200 µL/well PBS followed by blocking with 200 µL/well of cell culture medium for 1 hr at room temperature. The blocking solution was removed and replaced by 100 µL/well of a thoroughly washed ASC suspension containing 125 (6E2 and EBV-immortalized cell lines) or 250,000 cells (PBMC). The plates were then incubated for 24 hr at 37°C with 5% $CO_2$. Cell suspension was then removed, and the wells washed three times with 200 µL/well PBS, followed by the addition of 100 µL of peptide-tagged VAR2CSA antigens diluted in PBS 0.1% BSA (at 1 µg/mL) and incubation for 1 hr at room temperature. Plates were then washed again three times with PBS and 100 µL of fluorophore-conjugated detection reagents were added (anti-GAL, 490, anti-TRAP-550, Strep-Tactin-380, and anti-WASP-640), all diluted to 0.5 µg/mL in PBS 0.1% BSA. The frequency of total IgG-producing cells when using PBMC samples was determined using 25,000 pre-stimulated PBMC in two separate duplicate wells. To detect IgG-producing cells, 100 µL/well of biotinylated mAb anti-human IgG (3850-6, Mabtech) diluted to 0.5 µg/mL, were added followed by Strep-Tactin-550. After a final three washes with PBS, 50 µl/well of Fluorescent enhancer II (Mabtech) were added followed by incubation for 10 min at room temperature. Enhancer was then removed, and the underdrain of the plate was carefully removed. The wells were left to dry protected from light at room temperature. When the 6E2 mouse hybridoma cell line was tested, a blocking step prior to antigen addition was included, to block the coating goat anti-mouse antibody from binding the peptide detection antibodies. About 200 µL/well filtered mouse serum (Innovative Research, USA) diluted in PBS (1:5) was used for 2 hr at room temperature followed by three washes with 200 µL/well PBS. Data were then immediately acquired using the Mabtech IRIS reader. For each well, images in each detection channel (LED490, LED550, LED380, and LED640) were taken and number of spots per well and RSV were automatically counted using the Apex software version 1.1.7. Intensity thresholds for each detection channel were chosen (for each cell line/monoclonal antibody tested and for the PBMC samples) and consistently used throughout all the experiments. Percentages of single, double, triple, and quadruple positive spots were also calculated.

## Data analysis

Statistical analyses were performed using GraphPad Prism version 6 (GraphPad Software, La Jolla, CA). The Friedman test was used to detect differences in the mean across experimental groups followed by Dunn's multiple comparison test. An alpha of 0.05 was used.

# Acknowledgements

The research was supported by the Danish International Development Agency (DANIDA) grant MAVARECA-II (17-02-KU) and by the National Institute of Allergy and Infectious Diseases (NIAID) of the National Institutes of Health (NIH) under award number R21AI164147-01. The content is solely the responsibility of the authors and does not necessarily represent the offical views of the National Institutes of Health. NA was supported by a MAVARECA-II PhD studentship. MdPQ received funding from the European Union's Horizon 2020 research and innovation program under the Marie Skłodowska-Curie grant agreement No. 101028915. The authors thank Morten Agertoug Nielsen (University of Copenhagen, Denmark) for providing the 6E2 mouse hybridoma cell line, and Jonathan Renn and Patrick E Duffy (National Institute of Allergy and Infectious Diseases, NIH, USA) for the plasmids encoding the full-length VAR2CSA 3D7, M920, and Malayan Camp allelic variants.

# Additional information

### Competing interests

Peter Janhmatz, Niklas Ahlborg: is an employee of Mabtech AB, the manufacturer of the IRIS instrument and several of the analytical reagents used in the study. The other authors declare that no competing interests exist.

### Funding

| Funder | Grant reference number | Author |
| --- | --- | --- |
| Danish International Development Agency | MAVARECA-II (17-02-KU) | Lars Hviid |
| National Institute of Allergy and Infectious Diseases | R21AI164147-01 | Lars Hviid<br>Maria del Pilar Quintana |
| Horizon 2020 | Marie Sklodowska-Curie grant agreement 101028915 | Maria del Pilar Quintana |

The funders had no role in study design, data collection and interpretation, or the decision to submit the work for publication.

### Author contributions

Henriette Hoffmann-Veltung, Investigation, Validation, Visualization, Writing – review and editing; Nsoh Godwin Anabire, Investigation, Resources, Writing – review and editing; Michael Fokuo Ofori, Funding acquisition, Project administration, Resources, Supervision, Writing – review and editing; Peter Janhmatz, Niklas Ahlborg, Resources, Writing – review and editing; Lars Hviid, Conceptualization, Funding acquisition, Project administration, Supervision, Writing – original draft, Writing – review and editing; Maria del Pilar Quintana, Conceptualization, Formal analysis, Funding acquisition, Investigation, Methodology, Project administration, Supervision, Validation, Visualization, Writing – original draft, Writing – review and editing

### Author ORCIDs

Lars Hviid http://orcid.org/0000-0002-1698-4927
Maria del Pilar Quintana http://orcid.org/0000-0001-6190-3324

### Ethics

Human PBMC samples were collected after informed consent was obtained from the donors. Ethical approval for the study was obtained from the institutional review board of the Noguchi Memorial

Institute for Medical Research (NMIMR-IRB CPN 005/20-21) and the ethical review committee of the Ghana Health Service (GHS-ERC 005/08/20).

## Decision letter and Author response

Decision letter https://doi.org/10.7554/eLife.79245.sa1
Author response https://doi.org/10.7554/eLife.79245.sa2

## Additional files

### Supplementary files
- Supplementary file 1. List of primers used for VAR2CSA domain cloning.
- MDAR checklist

### Data availability
All data generated is presented in the main manuscript or in the figure supplements.

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
