## [Editor Report]

The information included in the manuscript attests to the intended rigor taken by the investigators in providing a high-quality resource material for the use by scientists interested in antibody-mediated responses to pathogens.

---

## [Decision Letter]

**Decision letter after peer review:**

Thank you for submitting your article "Analysis of allelic cross-reactivity of monoclonal IgG antibodies by a multiplexed reverse FluoroSpot assay" for consideration by *eLife*. Your article has been reviewed by 2 peer reviewers, and the evaluation has been overseen by a Reviewing Editor and Betty Diamond as the Senior Editor. The following individuals involved in review of your submission have agreed to reveal their identity: Justin YA Doritchamou (Reviewer #1); Adrian Luty (Reviewer #2).

Essential revisions:

1) The idea and use of "true" vs "apparent" cross-reactivity of antibody terms is a bit confusing and I don't think this assay is fine-tuned to clearly assess such specific epitopes.

2) Figure 1—figure supplement 2A: any comment on why Female pool antibody reactivity was a bit higher on NF54 ID1-ID2? Can this be related to pattern observed with the donors in Figure 5, figure supplement 1A?

3) Can the authors explain how the background fluorescence was corrected in the different channels, especially for LED 380?

4) Please provide the parity of the women donors to give more context to the observed pattern of antibody reactivity described in this work.

5) The authors could have included IT4 FL in the screening of donors PBMC as control even in a 1x1 format to provide information of general reactivity to VAR2CSA in comparison domain-specific reactivity.

6) Sentence in lines 70-71 is redundant to lines 54 – 56.

7) Discussion: Whether the discrepancy in 6E2 binding measured by ELISA vs FluoroSpot is due to changes in antigen display in both assays or affinity of antibody should be clearly discussed by the authors, especially regarding the binding of 6E2 to NF54 and M920 variants of ID1-ID2.

8) In the context of the figures, the schematics provided are very clear and instructive.

However, a rationale could be provided for presenting the data in Figure 5 as percentages as a function of the positive control (total IgG) condition, rather than as numbers of spots per well.

9) It seems that it is the authors' intention that all the Figure supplements provided should be included in the main body of the article. Could an effort be made o switch at least some of these supplements to 'true' supplementary data, since the article would otherwise be overburdened with figures and not all of them are essential to getting the relevant messages across.

---

## [Author Response]

Essential revisions:1) The idea and use of "true" vs "apparent" cross-reactivity of antibody terms is a bit confusing and I don't think this assay is fine-tuned to clearly assess such specific epitopes.

The literature is replete with reports claiming antibody cross-reactivity based on reactivity of immune sera with multiple allelic variants of polymorphic antigens (a recent and typical example can be found in Franca et al. Front Cell Infect Microbiol 11, 616230, 2021). The findings reported in such studies are generally consistent with the presence of truly cross-reactive antibodies, where individual antibodies are able to recognize multiple variants. However, they are equally consistent with the alternative scenario, where multiple antibody specificities, each able to recognize only one variant, exist side-by-side. Because each antibody only recognizes a single allelic variant in the second scenario, they are not cross-reactive, and the observed cross-reactivity is therefore apparent rather than real. The distinction is not merely academic. The former scenario predicts that vaccination with (or natural exposure to) a single allelic variant of the antigen would lead to broadly cross-reactive immunity against all the variants of that antigen. In marked contrast, the latter scenario predicts variant-specific immunity to only the variant the immune system is exposed to. Despite this important difference, the distinction between the two scenarios is very often ignored.

Distinguishing between the two requires analysis of antibody responses elicited by vaccination with a single variant, depletion of antibodies with a single variant (as one of the reviewers has elegantly described in a couple of studies), or analysis at the single B-cell level, as we have done in the present study. We therefore respectfully maintain that the distinction is important, even if – or perhaps especially because – it is so often ignored. Please note that we are discussing antibody cross-reactivity, not cross-reactive epitopes (an epitope that is conserved among several allelic variants of a given antigen, and therefore capable of eliciting truly cross-reactive antibodies).

With respect to the comment regarding fine-tuning to assess specific epitopes, we regret not to understand well what is meant. Our study was not designed to identify specific epitopes, and we make no claims to have identified cross-reactive epitopes.

2) Figure 1—figure supplement 2A: any comment on why Female pool antibody reactivity was a bit higher on NF54 ID1-ID2? Can this be related to pattern observed with the donors in Figure 5, figure supplement 1A?

We agree with the reviewer regarding this observation. We indeed observed a higher reactivity against the NF54 ID1-ID2 domain when a female serum pool from Ghanaian donors was tested. We however did not dwell too much on that finding since the only aim of these ELISA experiments was to make a general assessment and QC of the expressed antigens rather than to establish if a differential recognition of the variants used was present. The NF54 isolate being from Africa might better resemble the VAR2CSA variants expressed by the parasites previously infecting these women. This, however, is speculation and since we did not collect parasite samples from the same donors and/or sequence the VAR2CSA sequences expressed by the parasites infecting them, we cannot establish if sequence similarity explains the observation pointed out by the reviewer.

With respect to whether the pattern observed in ELISA is related to the pattern observed in the selected donor presented in Figure 5, figure supplement 1A, we do not think that the two can be easily correlated. First, because the serum pool used for ELISA does not match the donors used for the FluoroSpot assay. Secondly, because the donor presented (MK601) was one of the highest responders for this particular domain (NF54 ID1-ID2). We also observed a wide variation across the few donors we tested and in a couple of cases the highest number of spots was not for the NF54, but for the IT4 or M920 variants. To establish if NF54 is in general better recognized both in ELISA and in FluoroSpot the number of samples tested would have to be increased.

3) Can the authors explain how the background fluorescence was corrected in the different channels, especially for LED 380?

We used the Mabtech IRIS reader and the Apex software as described in the methods section for data acquisition. The advantage of the software is that raw images of the wells are automatically acquired without any external manipulation (no calibration or manual focusing is performed) and the only acquisition setting that can be changed is the spot intensity (for each channel), which acts as a threshold to include or exclude spots in the final count. There are also brightness and contrast sliders that only affect the visual aspect of the acquired images, but they do not affect the spot count or RSV values. We have added a paragraph (lines 450-452 in the “clean” manuscript) in the Materials and methods section briefly stating this.

4) Please provide the parity of the women donors to give more context to the observed pattern of antibody reactivity described in this work.

We have added this information in lines 215-216.

5) The authors could have included IT4 FL in the screening of donors PBMC as control even in a 1x1 format to provide information of general reactivity to VAR2CSA in comparison domain-specific reactivity.

We appreciate this comment and agree that this would have been interesting to test. Unfortunately, the IT4 FL recombinant antigen we had available did not have any of the necessary detection tags for FluoroSpot and therefore could not be tested even in a 1×1 configuration. However, as mentioned in the Discussion section, we fully agree that inclusion of FL constructs in future applications of the FluoroSpot would be very interesting to perform to see if we observe a higher level of cross-reactivity, in contrast to what we have preliminary observed here with the single domains. Indeed, we are planning such an investigation.

6) Sentence in lines 70-71 is redundant to lines 54 – 56.

We have modified lines 54-56 to remove the redundancy pointed out by the reviewer.

7) Discussion: Whether the discrepancy in 6E2 binding measured by ELISA vs FluoroSpot is due to changes in antigen display in both assays or affinity of antibody should be clearly discussed by the authors, especially regarding the binding of 6E2 to NF54 and M920 variants of ID1-ID2.

It is correct that the 6E2 reacted with IT4, M920, and NF54 allelic variants of VAR2CSA ID1-ID2, whereas it did not react with the NF54 allele in the FluoroSpot assay. While we do not have any formal evidence explaining this discrepancy, the most likely explanation that is consistent with our data is a lower sensitivity of the FluoroSpot assay compared to ELISA. This is explicitly stated in the original manuscript and is maintained in the revised manuscript (lines 181-183).

8) In the context of the figures, the schematics provided are very clear and instructive.However, a rationale could be provided for presenting the data in Figure 5 as percentages as a function of the positive control (total IgG) condition, rather than as numbers of spots per well.

When presenting data from several donors in the same panel (as in Figure 5), it is necessary to normalize the data, as different donors are highly likely to have different frequencies of ASC. In Figure 5—figure supplement 2 we present the data as spots per well, instead of as normalized percentages, because normalization is not needed in that figure, since the comparison is not between different donors but between the 4×4 and the 1×1 configurations, normalized to the same number of total ASC.

9) It seems that it is the authors' intention that all the Figure supplements provided should be included in the main body of the article. Could an effort be made o switch at least some of these supplements to 'true' supplementary data, since the article would otherwise be overburdened with figures and not all of them are essential to getting the relevant messages across.

We do not expect or prefer that all the figure supplements are included in the main body of the article. Rather, all figures labeled as supplements are intended as supplementary data and we expect that the main paper will only include the main figures (a total of five). For the version submitted to *eLife* we included them all in the compiled pdf document to facilitate the reviewing process.